# Tuning the Selectivity of LaNiO$_3$ Perovskites for CO$_2$ Hydrogenation through Potassium Substitution

**Constantine Tsounis, Yuan Wang** [†] [iD]**, Hamidreza Arandiyan** *,[‡] [iD]**, Roong Jien Wong** [§] [iD]**, Cui Ying Toe** [iD]**, Rose Amal and Jason Scott** *

Particles and Catalysis Research Group, School of Chemical Engineering, The University of New South Wales, Sydney, NSW 2052, Australia; c.tsounis@unsw.edu.au (C.T.); yuan.wang4@unsw.edu.au (Y.W.); roong.jien.wong@rmit.edu.au (R.J.W.); c.toe@unsw.edu.au (C.Y.T.); r.amal@unsw.edu.au (R.A.)

* Correspondence: hamid.arandiyan@sydney.edu.au (H.A.); jason.scott@unsw.edu.au (J.S.); Tel.: +61-2-9114-2199 (H.A.); +61-2-9385-7361 (J.S.)

† Permanent Address: School of Chemistry, The University of New South Wales, Sydney, NSW 2052, Australia.

‡ Permanent Address: Laboratory of Advanced Catalysis for Sustainability, School of Chemistry, The University of Sydney, Sydney, NSW 2006, Australia.

§ Permanent Address: Applied Chemistry and Environmental Science, School of Science, RMIT University, Melbourne, Victoria 3000, Australia.

**Abstract:** Herein, we demonstrate a method used to tune the selectivity of LaNiO$_3$ (LNO) perovskite catalysts through the substitution of La with K cations. LNO perovskites were synthesised using a simple sol-gel method, which exhibited 100% selectivity towards the methanation of CO$_2$ at all temperatures investigated. La cations were partially replaced by K cations to varying degrees via control of precursor metal concentration during synthesis. It was demonstrated that the reaction selectivity between CO$_2$ methanation and the reverse water gas shift (rWGS) could be tuned depending on the initial amount of K substituted. Tuning the selectivity (i.e., ratio of CH$_4$ and CO products) between these reactions has been shown to be beneficial for downstream hydrocarbon reforming, while valorizing waste CO$_2$. Spectroscopic and temperature-controlled desorption characterizations show that K incorporation on the catalyst surface decrease the stability of C-based intermediates, promoting the desorption of CO formed via the rWGS prior to methanation.

**Keywords:** selectivity tuning; CO$_2$ methanation; reverse water gas shift

## 1. Introduction

An economical yet effective removal and utilization of CO$_2$ has not been implemented on a wide scale level to reduce its impact on climate change through the greenhouse effect. This makes the efficient methanation of CO$_2$ or the production of CO through the reverse water gas shift (rWGS) reaction a promising solution if it can be implemented viably [1,2]. Current major solutions of CO$_2$ utilization revolve around the production of synthetic chemicals such as methanol, various carbonates, and urea. Combined with other carbon sequestration efforts, this only accounts for the removal of a minute portion of the ~ 30 gigatonnes (Gt) of CO$_2$ emitted annually [3–5]. The production of these liquid chemicals generally require intensive operating conditions, and reactions are considerably slower than the rWGS or Sabatier methanation reactions [1]. Furthermore, there is a significant global push towards realizing renewable hydrogen production at scale through the use of water electrolysis combined with renewable energy inputs [2]. This presents significant opportunity for the implementation of CO$_2$ hydrogenation through these routes, which are able to convert CO$_2$ into CO (via the rWGS) and CH$_4$ (via methanation).

CH$_4$ and CO produced from the thermal reduction of CO$_2$ can be used to create sustainably sourced synthetic fuels, as well as act as building blocks for other reactions which are able to produce value added products. These include aromatic compounds, alcohols, ketones, and carboxylic acids. Moreover, the process of creating these products can reduce anthropogenic CO$_2$ in the atmosphere, presenting itself as a potential multifaceted approach to reducing CO$_2$ pollution [4].

$$CO_2 + 4H_2 \rightarrow CH_4 + 2H_2O \qquad \text{Sabatier Reaction [6]} \qquad \Delta H = -165.0 \text{ kJ mol}^{-1}$$
$$CO_2 + H_2 \rightarrow CO + H_2O \qquad \text{Reverse Water Gas Shift [6]} \qquad \Delta H = 41.15 \text{ kJ mol}^{-1}$$

Further to this, there is a need for specific product ratios of CH$_4$ and CO, which are able to be used as reactant streams for further hydrocarbon reforming. It has been shown previously that the addition of CO during the dehydrocondensation of CH$_4$ into aromatic products over a zeolite-based catalyst was able to significantly improve catalytic stability and selectivity, resulting in an increase in catalytic performance [7–9]. Therefore, in order to increase the feasibility and applicability of utilizing the catalytic conversion of CO$_2$ into sustainable fuels and products, this work aimed to tune the selectivity of CO$_2$ reduction between the rWGS and methanation to specific product ratios, which can be used more efficiently for downstream processes. Perovskite-type oxides with the general formula ABO$_3$ containing both rare earth elements and 3d transition metals have received much attention in recent years [10–13]. Supported base metal oxides (e.g., cerium oxide [14] and zirconium oxide [15]) and perovskites such as LaMO$_3$ (M = Co, Ni, Mn) [16]) are catalytically active in CO$_2$ reduction at above 200 °C, while showing good selectivity towards the Sabatier reaction. Particularly, Ni as the B-site cation has been previously probed for CO$_2$ methanation. In using LaNiO$_3$ (LNO), which is reduced to Ni/La$_2$O$_3$ upon activation [17,18], the Ni species have been shown to be an active component in both Sabatier methanation and the rWGS routes, and, under various conditions, will have different reaction selectivity and activity [19,20]. However, very few works have been reported to exploit the strong thermal, hydrothermal, excellent redox properties, and flexible composition of the perovskite structure [12,21,22], and specifically manipulate selectivity away from CO$_2$ hydrogenation to the rWGS pathways, achieving an optimal ratio of CO and CH$_4$ in the product stream.

Alkali metals such as Li, Na, and K within various metal-based supported catalysts have been shown to promote rWGS routes due to their role of forming active sites, which promote formate species that decompose into CO. These active sites also alter the properties of neighboring species by potentially changing stability, particle size, and intermediate bond types [23–26]. Furthermore, K has been shown previously to be readily incorporated into La-based perovskites through the A-site substitution of La, and is able to change its electronic properties due to differences in ionic radius and preferred oxidation states [27,28].

In this work, we exploited the high selectivity of the LNO perovskite catalyst towards the Sabatier reaction and, by controlling potassium incorporation in the catalyst to form La$_{1-x}$K$_x$NiO$_3$ (x= 0, 0.1, 0.2, or 0.3, denoted as 0K LNO, 10K LNO, 20K LNO, or 30K LNO), tuned the reaction pathway toward the rWGS routes. The resulting changes in morphology, electronic properties, reducibility, and surface compositions emerging from the K substitution were analyzed in terms of their effect on the selectivity of the catalysts.

## 2. Results and Discussion

### 2.1. Material Characterization

The XRD pattern (Figure 1) for the LNO perovskite without K substitution showed high crystallinity and diffraction peaks that corresponded to the rhombohedral LaNiO$_3$ perovskite phase (JCPDS 00-034-1181). It was observed that after partial substitution of La with K, a secondary major phase appeared corresponding to the formation of NiO (JCPDS 01-078-0643) in conjunction with the crystalline LaNiO$_3$ structure. A similar phenomenon was obtained in other works with the La$_x$K$_{1-x}$CoO$_3$ (x = 0–0.3) perovskite, whereby K substitution induced the formation of the major B-cation oxide (Co$_3$O$_4$) [27].

It was postulated that, in our case, a structural defect occurred as a result of the different ionic radii between La$^{3+}$ and K$^+$, resulting in an imperfect substitution, where not all La$^{3+}$ ions were replaced by K$^+$ within the structural A-site. This may result in an excess of Ni species not incorporated into the perovskite structure, which react with oxygen to form NiO during calcination. The NiO species formed increased in abundance as K substitution increased, as seen in Figure 1 through the increasing XRD peak intensity of NiO. Slight peak broadening in the characteristic peak of the LNO perovskite also indicated lattice distortion, potentially due to the substitution of K cations into the structure. However, the presence of K species could not be detected in XRD analysis, suggesting that the K in the catalyst was in a well-dispersed form, such as nanocrystals [26].

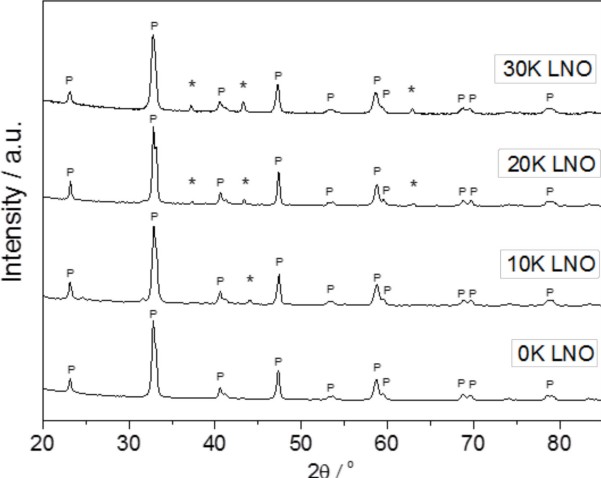

**Figure 1.** Characteristic peak intensity normalized XRD patterns of 0–30K LNO catalysts. The (*) symbol represents the NiO standard phase while (P) symbol represents the LaNiO$_3$ perovskite phase.

The Brunauer-Emmett-Teller (BET) surface area for all catalysts are shown in Table 1. The 0K LNO possessed the highest surface area value of 10.5 m$^2$ g$^{-1}$. Previous works reported surface areas of LaNiO$_3$ with various synthesis methods ranging from approximately 1–10 m$^2$ g$^{-1}$ [29,30]. For the 10–30K LNO samples, the BET surface area was lower following La substitution with the K ions. This was partially attributed to the distortion of the rhombohedral lattice structure of the single phase LaNiO$_3$ due to a change in radius of the A-site cation, and has also been correlated to the formation of low melting point K compounds, as shown by Xu and coworkers [27], which were deformed during the calcination.

**Table 1.** Surface parameters of the 0–30K LNO samples.

| Catalyst | BET Surface Area (m$^2$ g$^{-1}$) | K Surface Amount [a,b] (at. %) | Overall K Amount [b,c] (at. %) |
|---|---|---|---|
| 0K LNO | 11 | 0 | 0 |
| 10K LNO | 6.5 | 30.4 | 0.56 |
| 20K LNO | 5.2 | 23.8 | 1.3 |
| 30K LNO | 7.1 | 3.56 | 2.4 |

[a] Determined using XPS surface analysis. [b] Calculated based on percentage of overall metal content. [c] Calculated through ICP-OES analysis.

N$_2$ adsorption and desorption curves showed a small hysteresis indicating adsorbed N$_2$ on the nanoparticle surface for the 0K LNO sample, which indicated the limited presence of meso or micropores [31], as can be seen in Figure S1. These pores are beneficial to the overall catalytic performance as they allow for higher adsorbed surface oxygen capabilities and better redox properties. The hysteresis appeared to decrease as K was incorporated into the catalyst structure, providing further

evidence of distortion within the crystal lattice due to the K substitution amount. Interestingly, Table 1 shows that as higher amounts of K were incorporated during synthesis there was a decrease in K surface species, suggesting that the substitution effect was more evident for higher K amounts in our system.

For reductive type reactions, catalyst reducibility is a good indicator of the redox properties and abilities over a temperature range in which the catalyst is active. Particularly for metal oxide-based catalysts, the formation and abundance of active sites is a driving force for the catalytic reduction of $CO_2$, as well as a factor defining reaction pathway [17]. Figure 2 presents the reductive potential through $H_2$ temperature programmed reduction ($H_2$ TPR) of the 0–30 K LNO catalysts. The results show two major reduction zones across a temperature range of approximately 250–575 °C. Singh et al. described two possible reaction paths for $LaNiO_3$ reduction, which were distinguishable through the relative sizes of the first ($\alpha$) and second ($\beta$) reduction peak. However, both pathways eventually led to solid phase crystallization, forming Ni supported on $La_2O_3$ [17]. For the crystalline 0K LNO, whereby the only phase present exists as a rhombohedral $LaNiO_3$ perovskite, it can be seen that the $\beta$ peak is approximately two times the size of the $\alpha$ peak. The pathway likely followed a two-step method, whereby the $\alpha$ peak indicates the reduction of $Ni^{3+}$ to $Ni^{2+}$, and the $\beta$ peak indicates the exsolution of Ni onto the perovskite surface through the reduction of $Ni^{2+}$ to $Ni^0$, forming an active Ni site, which has been shown to participate in $CO_2$ methanation routes [19,32].

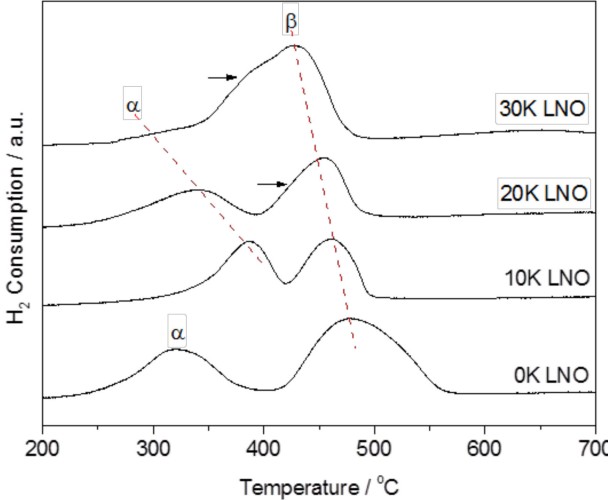

**Figure 2.** $H_2$ TPR profiles of the 0–30K LNO samples.

In the case of 10–30K LNO, a similar reduction pathway was proposed. However, slight changes in the peak profiles were seen, likely as a result of an increase in structural defects as K was introduced. It can be seen that the $\alpha$ peak initially shifted toward higher temperatures, the extent of which may be directly influenced by the amount of K surface-based species. The effect tended to reduce for the 20–30K LNO samples as the peaks shifted back towards lower temperatures. This suggests that surface K species may increase the stability of $Ni^{3+}$, as seen by their increased resistance to reducibility [17]. Conversely, the $Ni^{2+}$ to $Ni^0$ reduction peak shifted slightly to lower temperatures as K concentrations in the bulk catalyst increased, which may result from perovskite lattice distortion due to changes in the A-site ionic radius. Additionally, a tailing peak, which developed on the left side of the $\beta$ peak in the 20K LNO sample, increasing into a shoulder on the 30K LNO sample, may be ascribed to the reduction of NiO, which may not have entered the perovskite after substitution. The increasing NiO crystal size indicated by the proportionally larger NiO XRD diffraction peaks (Figure 1) may create a slightly more easily reduced $Ni^{2+}$ phase, which can be seen by the increasing shoulder peak at lower temperatures in these samples. The decreasing area of the $\alpha$ and $\beta$ peaks in the 10–30K LNO samples correlated to the presence of less reducible Ni species once K was incorporated into the structure and surface. This

may be ascribed to K hindering the abundance of active Ni species that participate in redox reactions, and possibly influenced hydrogen activation on the catalyst [20].

TEM images of the 20K LNO sample shown in Figure 3a–c reveal NiO particles were formed on the perovskite surface during synthesis. There appears to be an area of high contrast surrounding the Ni species, which may possibly be ascribed to K species on the catalyst surface, which are likely in oxide form [33]. EDS mapping of the 20K LNO sample, as shown in Figure 3d–h, further confirms the presence of surface K species, in conjunction with La, Ni, and O species also present on the catalyst surface. When comparing 0K LNO and 20K LNO after the reduction treatment, prior to performance evaluation, TEM images demonstrated a significant increase in particle size and decrease in uniform distribution of reduced Ni nanoparticles in the 20K LNO sample relative to the 0K LNO sample (Figure S2). This effect was likely due to the reduction of bulky NiO species previously formed on the 20K LNO surface during synthesis, as La was replaced by K cations.

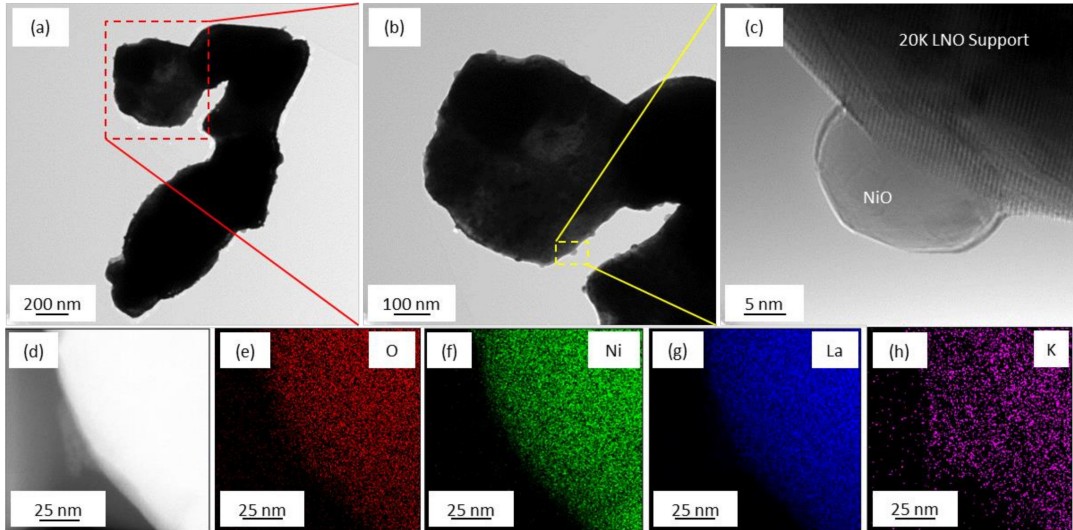

**Figure 3.** (**a**–**c**) TEM images of the fresh 20K LNO catalyst and (**d**–**h**) EDS mapping of O, Ni, La, and K on the surface of the sample.

XPS analysis was used to analyze the changing chemical properties of the samples as K ions were substituted into the perovskite structure. Figure 4a depicts two main deconvoluted peaks of K, the first of which at approximately 292.6 eV corresponded to the $2p_{3/2}$ orbital, with an accompanying $2p_{1/2}$ orbital peak at approximately 295.4 eV. These peaks were shown to correspond to a K–O group, which, in our case, likely consisted of $K_2O$ which formed through the calcination of K surface species in air [33–35]. Observation showed a shift to higher binding energies for both of these peaks to 293.0 eV and 295.8 eV in the 30K LNO sample for the $2p_{3/2}$ and $2p_{1/2}$ orbitals, respectively, indicating oxidation of the surface K cation sites as more K was substituted into the perovskite structure. Furthermore, the decreasing intensity of these peaks in the 30K LNO sample was a consequence of the decreasing amount of surface K, shown in Table 1.

For the Ni 2p orbitals shown in Figure 4b, the 0K LNO sample exhibited two main peaks at approximately 855.9 eV and 873.0 eV, which were previously reported to correspond to the $2p_{3/2}$ and $2p_{1/2}$ orbitals of $Ni^{3+}$ in the perovskite structure, respectively [36]. It can be seen that these peaks shifted towards lower binding energies in the 10K and 20K LNO samples, indicating that Ni species were being reduced, the electron source of which likely originated from both surface and bulk K species. Further evidence of the reduction effect was seen through a shoulder peak at approximately 853.9 eV in the 0K LNO sample, which increased in intensity for the 10K and 20K LNO samples, indicative of the formation of $Ni^{2+}$ species. The shoulder, however, became much more pronounced, forming a full peak in the 30K LNO sample, confirming the presence of the $Ni^{2+}$ species facilitated by electron

donation from K–O species. Upon reduction during the reaction, it was seen that all $Ni^{3+}$ species were fully reduced to $Ni^0$, which as previously mentioned, formed an active species for $CO_2$ hydrogenation (Figure S3 (reduced sample Ni 2p XPS spectra)).

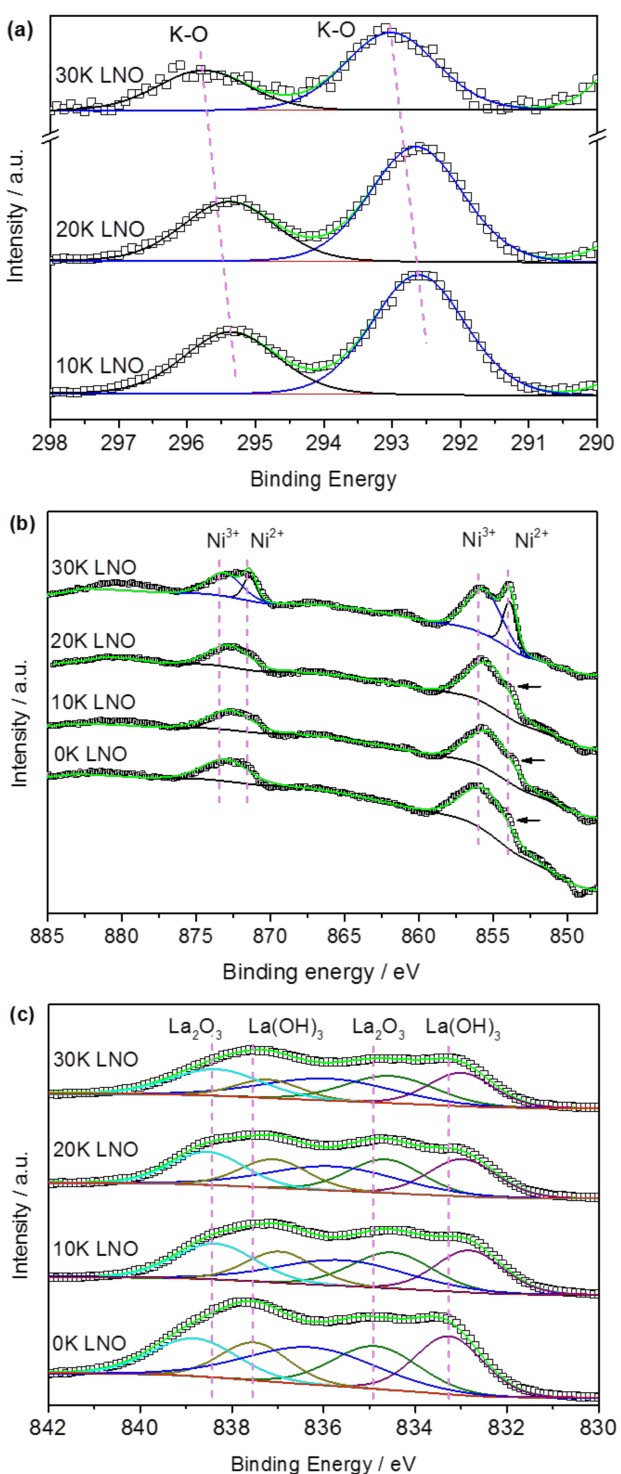

**Figure 4.** XPS profiles of (**a**) K 2p, (**b**) Ni 2p, and (**c**) La 3d of 0–30 K LNO samples.

The La 3d region was also examined and deconvoluted into two main doublets corresponding to $La_2O_3$ and $La(OH)_3$, whose spectra exhibited both core and satellite peaks. Peak overlap between $La(OH)_3$ and $La_2(CO_3)_3$ indicated that there was also $La_2(CO_3)_3$ present when considering carbonate

species present in the C1s spectra (Figure S4). Figure 4c shows the deconvolution of the La $3d_{5/2}$ orbital where the $La_2O_3$ core level and satellite peak were located at 833.2 eV and 837.5 eV ($\Delta E$ = 4.3 eV), while $La(OH)_3$ core level and satellite peaks were found to be at 834.9 eV and 838.9 eV ($\Delta E$ = 4.0 eV), respectively [37]. The peaks were observed to shift towards lower binding energies; however, as the amount of K substitution increased, the peaks drifted back towards slightly higher energies. It was proposed that the electron source of the $La_2O_3$ and $La(OH)_3$ reduction originated primarily from surface rather than bulk K species, and, therefore, the effect slightly decreased as the surface K concentration reduced to 3.56 at. % in the 30K LNO sample. Substituted K, in contrast to its effects on Ni, is believed to have minimal or negligible effect on the La species binding energies (electron densities) as K within the perovskite structure were not within the first shell vicinity of La, i.e., not having direct interaction with La. Despite the slight drift back to higher binding energies, the core level binding energies for $La_2O_3$ and $La(OH)_3$ in the 30K LNO sample still decreased overall by 0.4 eV and 0.3 eV, respectively, compared to 0K LNO, and may have effected intermediate bond stability during the reaction.

## 2.2. Catalytic Performance

### 2.2.1. Carbon Dioxide Conversion and Reaction Selectivity

The catalytic activity was observed at atmospheric pressure, over a temperature range that yielded from 0% $CO_2$ conversion to 100%, based on catalysts which were reduced in a flow of $H_2$ (25 mL min$^{-1}$) at 500 °C for 2 h. The reduction method was used to promote the formation of $Ni^0$, as $Ni^0$ is known to be an active species in both methanation and rWGS reactions [19,20]. The results from the catalytic activity test are shown in Figure 5 where $CO_2$ conversion is displayed as a function of temperature (light off curves). The highest performing catalyst, 0K LNO, exhibited $CO_2$ conversion with $T_{50\%}$ (temperature at which 50% of $CO_2$ is converted) of 240 °C, and $T_{100\%}$ at 270 °C, a performance comparable to some of the highest performing catalysts for the methanation reaction under similar reaction conditions [1]. For the K-substituted LNO catalysts, a trend in decreasing $CO_2$ conversion at a given temperature was identified and can be attributed to the incorporation of K. For the 10K, 20K, and 30K LNO catalysts, the $T_{50\%}$ temperatures are 290 °C, 306 °C, and 316 °C, respectively. The decrease in catalytic activity can be explained as a result of lesser stable adsorbed $CO_2$ on surface K, (shown to be in an intermediate $K_2CO_3$ state under in situ conditions), and the hindrance of activated H spill over from Ni species to the adsorbed $CO_2$ that K species may cause [33]. Furthermore, when comparing 10K LNO and 30K LNO in Figure 4c, where fewer electrons from K are transferred to La sites, the relatively lower electron density may decrease $CO_2$ adsorption and H activation rates on the catalyst. This effect, combined with a slight decrease in surface area shown in Table 1, may further lower overall catalytic activity. Moreover, the poor Ni distribution in the K-substituted samples and increased particle size (Figure S2) may also contribute to their decrease in $CO_2$ conversion rates [38]. Despite this, stability tests showed that for both 0K LNO and 20K LNO catalysts, over a period of 5 h, reaction activity and selectivity remained constant (Figure S5).

Reaction selectivity for all catalysts was observed to remain between the methanation of $CO_2$ ($CH_4$ production) and the reverse water gas shift (CO production), as seen in Figure 6. The 0K LNO can be seen as a high performing catalyst for the Sabatier methanation reaction, with 100% selectivity towards the methanation reaction for all temperatures. The increasing incorporation of K species within the perovskite structure can be used to tune the selectivity from 100% methanation at a given temperature, to also include CO in the product stream depending on catalyst and operating temperature, as seen in Figure 6. The reaction tuning can also be seen in Figure 7, which demonstrated the selectivity of all catalysts when overall $CO_2$ conversion was at 50%, showing that the changes in selectivity as a result of the addition of K were independent of overall conversion between the catalysts.

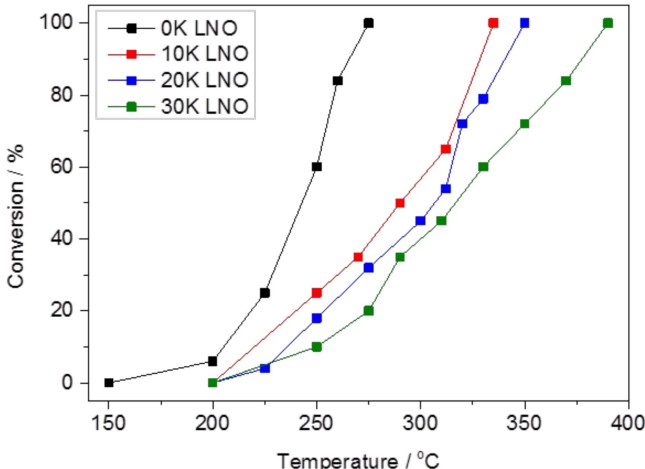

**Figure 5.** $CO_2$ conversion as a function of temperature for the 0–30K LNO samples.

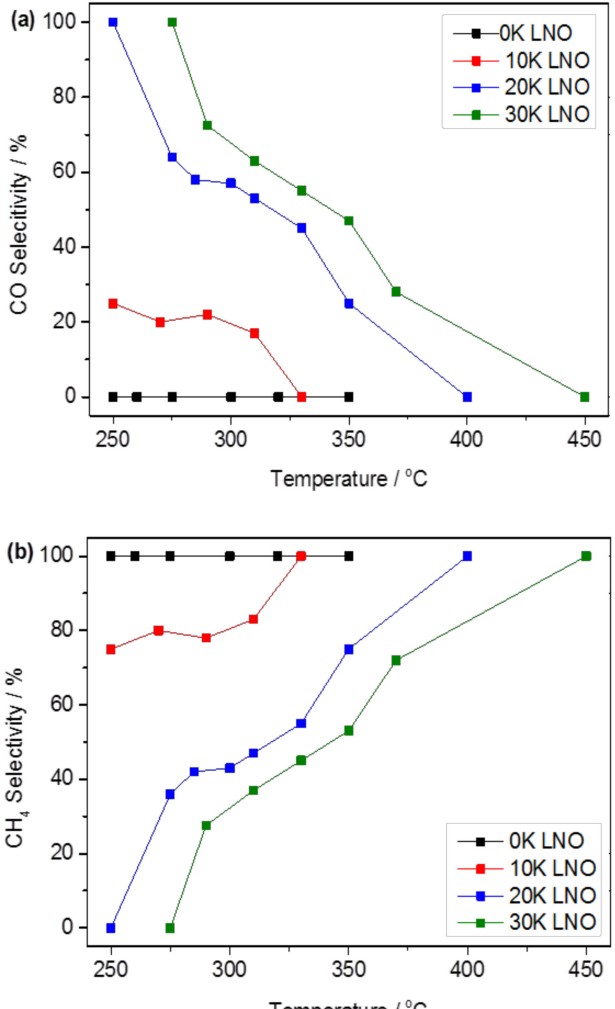

**Figure 6.** (**a**) Reaction selectivity towards the rWGS reaction and (**b**) reaction selectivity towards Sabatier methanation for all samples for 0–100% conversion.

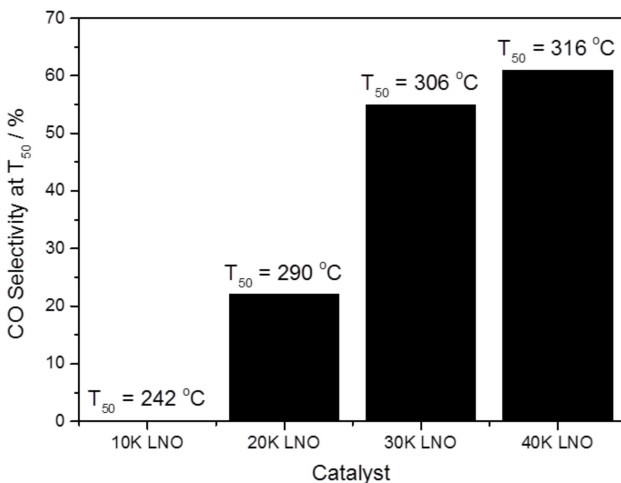

**Figure 7.** Selectivity towards the rWGS at $T_{50\%}$.

The selectivity of transition metal catalysts is largely influenced by the metal-support interaction, as well as the adsorption and stability of intermediate species on the support, which determine the most thermodynamically favorable reaction pathway [39]. For example, it has been reported that Ni/Ce-Zr-O-supported catalysts can exhibit up to 100% selectivity towards the rWGS reaction pathway [40], while $Ni/CeO_2$ can attain 100% selectivity towards $CO_2$ methanation [41]. Other transition metal catalysts also show a similar phenomenon with varying reaction selectivity [42–44]. Therefore, it is likely that, by incorporating K species, support interactions with the Ni active sites as well as the intermediate species stability on the support can be systematically tuned. This apparent shift in selectivity in K-substituted samples relative to 0K LNO may be partially contributed to by the increase in size of Ni particles (Figure S2). However, to fully understand the reasons for selectivity towards the rWGS, investigation into the role of K substitution was undertaken for the 0K LNO and 20K LNO samples.

### 2.2.2. Role of Potassium Substitution in Changing Reaction Selectivity

CO temperature programmed desorption (CO-TPD) was undertaken to understand the nature of CO adsorption/desorption equilibrium on the catalyst surface. Figure 8a outlines the CO desorption results of the 0K and 20K samples. It is clear that there were a number of CO desorption features that were identifiable for the two samples. The first peak for both samples (labelled as α at 264 °C in 0K LNO and 164 °C in 20K LNO) corresponded to CO species adsorbed on $Ni^{2+}$ [45]. It is likely that surface adsorption also occurred on the $K_2O$ surface species for 20K LNO, forming a range of intermediates such as carbonates and hydroxycarbonates, resulting in the shift of the first CO desorption peak to a lower temperature with lower intensity, indicating reduced CO uptake and stability. A second peak, present clearly only for 0K LNO at 535 °C (labelled as β), may be indicative of CO species adsorbed on $Ni^0$ particles [45]. Although the formation of Ni carbonyl complexes may also occur, these complexes are easily removed at low temperature <100 °C and do not form a significant part of the desorption profile [46]. The lack of obvious peak in this region for the 20K LNO sample was further evidence of the poorer stability and interaction of adsorbed CO on this sample. Two sharp desorption peaks in the 20K LNO sample were present in the region of 600 °C to 900 °C. These peaks may derive from the thermal decomposition of unstable $K_2CO_3$ species and have relatively high intensity due to the presence of other decomposition products, such as $CO_2$, KOH, and $KHCO_3$ [47].

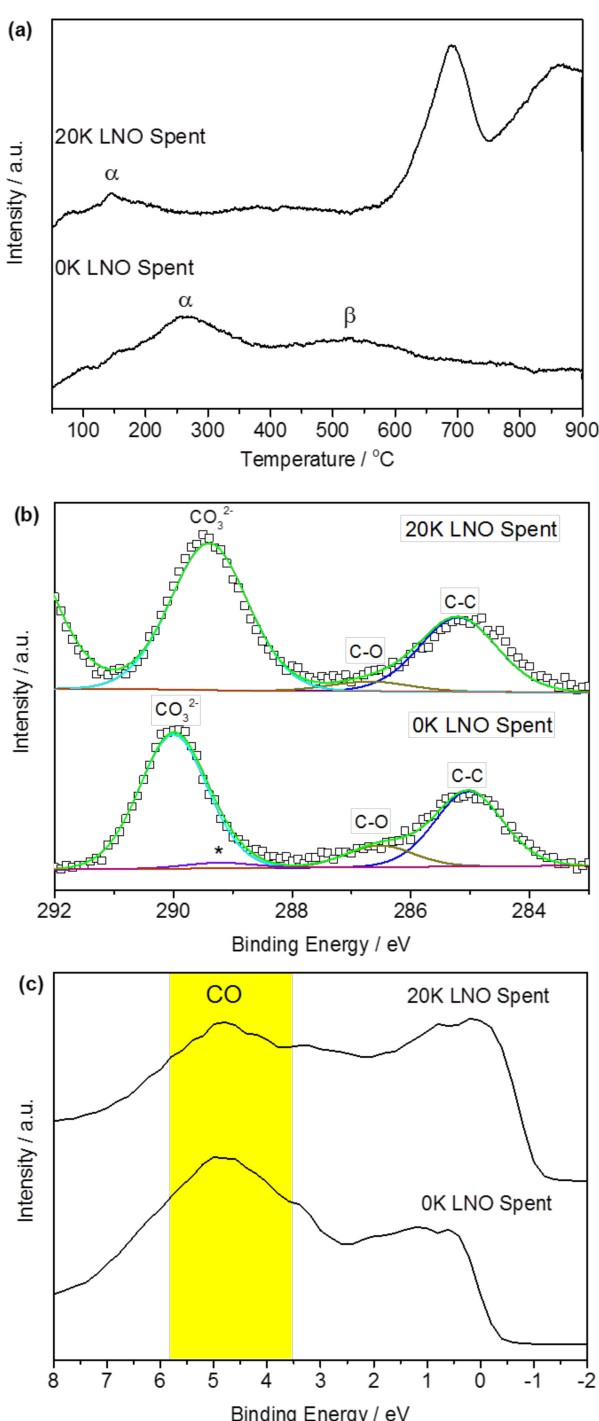

**Figure 8.** (**a**) CO-TPD profiles of the 0K and 20K LNO samples. (**b**) C1s spectra of the spent 0K and 20K LNO samples after a 2-h reduction step and 5-h reaction period at 300 °C. (**c**) XPS valence band spectra of the same spent samples.

In the presence of $K_2O$, it was shown by Maneerung et al. that $CO_2$ adsorption may take place on $K_2O$ surface species, leading to the formation of $K_2CO_3$ [33]. This intermediate was shown to be inherently more unstable when compared to $La_2O_2CO_3$ intermediates, which form as $CO_2$ is adsorbed on the La surface species. Therefore, when taken with the CO-TPD, it was proposed that the K incorporation is more likely to promote the formation of less stable C-based intermediates, which may preferentially desorb when compared to C-based intermediates in the analogous 0K LNO sample. Traditionally, one of the $CO_2$ methanation mechanisms has been attributed to $CO_2$ adsorption and

dissociation to form CO (via the rWGS), followed by CO hydrogenation [19,48,49]. It is possible that, in the presence of K species on the surface of our catalysts, these CO intermediates avoided complete hydrogenation before desorbing from the surface, forming CO instead of $CH_4$. Consequently, K incorporation and the associated surface species were able to tune reaction selectivity when comparing pristine LNO and K-substituted LNO samples, as seen in Figure 7.

Further evidence of C-based intermediate stability can be seen in Figure 8b, which provides the C1s XPS spectra of the spent 20K and 0K LNO samples, after a two-hour reduction in $H_2$ and five-hour reaction step. A peak with binding energy 289.2 eV (indicated with an asterisk) can be observed in the 0K LNO sample, corresponding to the formation of C=O species adsorbed on the catalyst surface [50]. In the case of the 20K LNO sample, the absence of this peak may indicate the evacuation of carbonate species during the reaction due to instability induced by the K cations forming carbonates or hydroxycarbonates, indicative of the shifting reaction selectivity of the K-substituted samples through promotion of rWGS routes. A similar phenomenon was observed for the C-O peak at 286.5 eV [50], which decreased in intensity with K incorporation (20K LNO), further alluding to the decrease in carbon-oxide species on the 20K LNO sample due to their increased instability during reaction.

In support of the role of K in shifting the reaction selectivity towards rWGS routes due to premature desorption of CO intermediates, analysis of the valence band spectra was undertaken in the spent samples, as shown in Figure 8c. It is apparent that at ca. 5 eV there was an increase in peak intensity for the 0K LNO when compared to the 20K LNO spent samples. This can be attributed to the presence of CO adsorbed on the catalyst surface due to overlapping of the CO valence band with the overall valence electronic structure of the catalyst surface, indicative of the heightened C-intermediate stabilities in the absence of K surface species [51]. This effect is demonstrated in Figure S6, which illustrates a proposed mechanism in which $K_2O$ species promote the instability of C-based intermediates, based on previous studies [33].

XPS analysis of the spent catalysts (Figure S7) showed the binding energy of the K 2p peak in the 20K sample to be at 292.9 eV with a split-orbit peak at 295.6 eV, compared to 292.6 eV and 295.4 eV, respectively, in the fresh 20K LNO sample. This indicates that as Ni species were reduced, K species were further oxidized, suggesting that K played the role of an electron source for this reduction step. Consequently, the oxidation state of K may be an underlying reason behind the shifting selectivity due to changes in intermediate stability [52]. The proposed K to Ni electronic pathway agrees with the work function-driven electron transfer mechanism observed in our earlier works on bimetallic alloys [53,54], resulting in the decreased $H_2$ TPR peak for the reduction of $Ni^{2+}$ into $Ni^0$.

## 3. Materials and Methods

### 3.1. Catalyst Preparation

A citrate method similar to that described by Irusta et al. [55] was used to create the nominal chemical compositions of $LaNiO_3$, $La_{0.9}K_{0.1}NiO_3$, $La_{0.8}K_{0.2}NiO_3$, and $La_{0.7}K_{0.3}NiO_3$ catalyst samples. The samples were denoted as 0K LNO, 10K LNO, 20K LNO, and 30K LNO, respectively. Each of the metal salts, $La(NO_3)_3 \cdot 6H_2O$ (>99%, Sigma-Aldrich®, St. Louis, MO, USA), $Ni(NO_3)_2 \cdot 6H_2O$ (>99%, Ajax Finechem, Scoresby, VIC, Australia), and $KNO_3$ (>99%, Sigma-Aldrich®) were dissolved in stoichiometric amounts in 4 mL of distilled water (Milli-Q, 18 m$\Omega$ cm). Citric acid (>99%, Sigma Aldrich ®) was added in a molar ratio such that for every one mole of the metal ion dissolved in solution two moles of citric acid were added. The dissolved metal ion solution was magnetically stirred for one hour at ambient temperature, forming a homogenous mixture. The solution was dried at 120 °C to give a porous gel, which was crushed to obtain a homogenous powder. The as-prepared solid was calcined in air for 3 h at 750 °C, being heated at a rate of 5 °C min$^{-1}$.

### 3.2. Catalyst Characterization

The crystallinity of each sample was analyzed by X-ray diffraction (XRD) using a Philips PANalytical Scherrer Diffractometer ($\lambda$= 0.154 nm, Amsterdam, Netherlands). Scattering intensity was recorded at a range of $8° < 2\theta < 90°$ with a $2\theta$ step of $0.03°$ and a time of 2 s per step. Hydrogen temperature programmed reduction ($H_2$ TPR) was performed using a Micromeritic Autochem II 2920 (Norcross, GA, USA) equipped with a thermal conductivity detector (TCD). Approximately 50 mg of the sample were first heated to 150 °C at a rate of 10 °C $min^{-1}$ in 40 mL $min^{-1}$ of argon, and held for 30 min to remove moisture. The temperature was then reduced to 50 °C, where the reducing gas was introduced at 40 mL $min^{-1}$ (10% $H_2$ in Ar). The sample was then heated to 900 °C at 10 °C $min^{-1}$, while $H_2$ consumed passing through the catalyst bed was measured through the TCD. CO temperature programmed desorption (CO-TPD) was also undertaken on the same apparatus. 50 mg of the fresh samples were initially treated with 30 mL $min^{-1}$ of He where it was heated to 150 °C, at a rate of 10 °C $min^{-1}$, and held for 30 min to remove moisture. The sample was then cooled to 50 °C where CO was introduced at a flow rate of 20 mL $min^{-1}$ for 30 min, followed by He at a flow rate of 20 mL $min^{-1}$ for 60 min. The temperature was then increased to 900 °C at a rate of 10 °C $min^{-1}$ while any desorbed CO in the effluent gas was detected through the TCD. BET surface area, pore volume, and pore size of the samples were measured using $N_2$ physisorption on a Micrometric Tristar 3030. Samples were pretreated under vacuum at 150 °C for 3 h to remove impurities. X-ray photoelectron spectroscopy (XPS) analysis was undertaken using a Thermo Fisher model ESCALAB250Xi (Waltham, MA, USA) and was used to probe the chemical states of various surface species with a wave energy (Al K$\alpha$) of 1486.68 eV and C1s reference of 284.8 eV. For the spectra of the spent samples, the samples were kept away from ambient oxygen by flushing the reactor with $N_2$, prior to storage in sealed $N_2$-purged vials before measurment. Morphology of the fresh catalyst samples was characterized using field-emission high resolution transmission electron microscopy (FE-HRTEM) on a Philips CM200 microscope (Amsterdam, Netherlands). Surface composition was analyzed using a high-angle annular dark-field scanning transmission electron microscope energy-dispersive X-ray spectroscopy (HAADF-STEM-EDS) with a JEOL F200 STEM (Tokyo, Japan). Chemical composition of the catalysts was determined through inductively coupled plasma optical emission spectrometry (ICP-OES) analysis and was undertaken on fresh samples that were digested in 1 part $HNO_3$ (70% w/w) and 3 parts HCl (36% w/w).

### 3.3. Catalytic Apparatus and Reaction

The performance of the catalysts was evaluated on a rig consisting of a quartz tube (6 mm inner diameter) acting as a fixed-bed micro reactor (see Figure S8 for a detailed schematic). Then, 50 mg of the sample was loaded on a bed of quartz wool. The samples were first reduced for 2 h at 500 °C in a 25 mL $min^{-1}$ stream of $H_2$ gas. The temperature was then lowered to 200 °C where $N_2$ at 13 mL $min^{-1}$ and $CO_2$ at 2 mL $min^{-1}$ were introduced and stabilized for 45 min, giving a total gas hourly space velocity (GHSV) of 48,000 mL (g h)$^{-1}$. The reactant gas was passed through the catalyst, with the exiting gas passing through a gas chromatograph (Young Lin-6100, Gyeonggi-do, Korea) containing a Carboxen-1010 PLOT column, equipped with a thermal conductivity detector (TCD). The outlet gas composition was analyzed over a temperature range which allowed for 0–100% $CO_2$ conversion of all catalysts.

## 4. Conclusions

The findings demonstrated the capacity for facile tuning of the selectivity in high-performing $LaNiO_3$ methanation catalysts towards the rWGS reaction. Tunable selectivity can potentially be beneficial for downstream hydrocarbon reforming while valorizining $CO_2$, providing a clear motivation for this method of rational catalyst design. Incorporating K into the catalytic structure produced K surface species, which were proposed to be the main contributor to the effect, by promoting the formation of thermally unstable C-intermediates, which potentially escape hydrogenation and instead

dissociate via the rWGS. The effect was demonstrated by comparing CO interaction with the catalysts, whereby 20K LNO samples showed less CO adsorption uptake and increased instability. Analysis of spent catalysts alluded to the presence of C-O and C=O based species, which had desorbed from the K-substituted samples while remaining for the 0K LNO samples, further evidence of the increased instability of C-intermediates in K-substituted samples. The results were confirmed by analyzing the valence band spectra of the samples, which showed a clear increase in binding energy intensity at 5 eV for the 0K LNO sample, corresponding to the presence of CO residue.

**Supplementary Materials:** The following are available online at http://www.mdpi.com/2073-4344/10/4/409/s1. Figure S1: $N_2$ adsorption and desorption isotherms. Figure S2: Reduced sample TEM images. Figure S3: Spent sample Ni 2p XPS spectra. Figure S4: C1s XPS spectra. Figure S5: Catalyst activity and selectivity stability tests. Figure S6: Proposed catalytic mechanism on K-substituted LNO samples. Figure S7: Spent sample K 2p XPS spectra. Figure S8: Process flow diagram of reactor setup used for the catalytic evaluation.

**Author Contributions:** C.T. undertook the experiments, analysed the results, and wrote the manuscript. H.A., J.S., and R.A. conceived the experiments and assisted with the results analyses. Y.W., C.Y.T., and, R.J.W. assisted with experimental work, results analyses, and writing the manuscript. All authors have read and agreed to the published version of the manuscript.

**Funding:** This research was funded by the Australian Research Council under the Laureate Fellowship Scheme [FL140100081].

**Acknowledgments:** The authors acknowledge the use of facilities within the UNSW Mark Wainwright Analytical Centre.

**Conflicts of Interest:** The authors declare no conflict of interest.

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
