# Peer review of "Tuning the Selectivity of LaNiO3 Perovskites for CO2 Hydrogenation through Potassium Substitution"

_catalysts, doi:10.3390/catal10040409_

Round 1

Reviewer 1 Report

Dear Authors,

The Authors define the purpose of the work as "in order to increase the feasibility and applicability of utilising the catalytic conversion of CO2 into sustainable fuels and products, this work aims to tune the selectivity  of CO2 reduction between the rWGS and methanation to specific product ratios, which can be used  more efficiently for downstream processes. " At the same time, in the first sentence of Chapter 1, Introduction, they write "An economical yet effective removal and utilisation of CO2 has not been implemented on a wide scale level to reduce its impact on climate change through the greenhouse effect." It is in this statement that one should look for the purposefulness - or rather the lack of it - of conducting research such as those in the presented work. Research should support the economy and apply in industry, not being just book knowledge. Laboratory tests as a first research step are important and undeniably necessary, but their results must be able to be realistically translated into their use, and not just remain in the form of a report on the shelf. Considering the essence of the research activities, described in the paper as the possibility of using a catalyst (developed by the Authors) for CO2 reduction in the direction of CO and CO2 methanation, they are of importance only in the cognitive aspect.          The basis for the implementation of process-oriented technologies, for which the Authors tested LaNiO3 perovskite catalysts through the substitution of La with K cations, is the production of hydrogen economically and environmentally justified. Hydrogen extraction in the process of water electrolysis (which is currently proposed) is highly energy-consuming (and therefore non-ecological), and thus uneconomical in the context of the use of hydrogen for the production of fuels by CO2 hydrogenation.                                                                                                                                                                                                                         For this reason, the development and testing of catalysts for processes that do not promise industrial use, is merely "art for art's sake."

The paper is written in a vague and chaotic way. It reads and analyzes with great difficulty.

The very title of Chapter 2 Results and Discussion raises doubts as to the chronology of individual chapters. And it really is so. There is no description of the experiment yet, it is not known how the catalyst was prepared, and the authors present the results of determining its properties (2.1. Material Characterization). Throughout this chapter, where the results are presented in the form of graphs and tables, the types of analytical equipment and analysis parameters used are not given. Only Chapter 3 is the descriptions of Materials and Methods.

In chapter 3.2. Catalyst Characterization, the last sentence should state the concentration of HNO3 and HCl used.

Chapter 3.3.  Catalytic Apparatus and Reaction lacks the layout of the laboratory stand, where the study was conducted. The position should be schematically represented and clearly marked, where the catalyst bed is located. The rule is to provide such a description of the experiment, so that it can be repeated with the possibility to verify the results. The tests were carried out only with one set of process parameters. The experiment should be conducted under different reaction conditions (using different process parameters) to optimize the process.

I find subsections 2.2.1. Carbon Dioxide Conversion and Reaction Selectivity and 2.2.2. as the most valuable . Role of Potassium Substitution in Changing Reaction Selectivity. I raise no objections to these chapters. They are described correctly with a developed scientific context.

However, in its current form, I do not recommend the article to be published.

Author Response

Reviewer 1

The Authors define the purpose of the work as "in order to increase the feasibility and applicability of utilising the catalytic conversion of CO2 into sustainable fuels and products, this work aims to tune the selectivity  of CO2 reduction between the rWGS and methanation to specific product ratios, which can be used  more efficiently for downstream processes. " At the same time, in the first sentence of Chapter 1, Introduction, they write "An economical yet effective removal and utilisation of CO2 has not been implemented on a wide scale level to reduce its impact on climate change through the greenhouse effect." It is in this statement that one should look for the purposefulness - or rather the lack of it - of conducting research such as those in the presented work. Research should support the economy and apply in industry, not being just book knowledge. Laboratory tests as a first research step are important and undeniably necessary, but their results must be able to be realistically translated into their use, and not just remain in the form of a report on the shelf. Considering the essence of the research activities, described in the paper as the possibility of using a catalyst (developed by the Authors) for CO2 reduction in the direction of CO and CO2 methanation, they are of importance only in the cognitive aspect.          The basis for the implementation of process-oriented technologies, for which the Authors tested LaNiO3 perovskite catalysts through the substitution of La with K cations, is the production of hydrogen economically and environmentally justified. Hydrogen extraction in the process of water electrolysis (which is currently proposed) is highly energy-consuming (and therefore non-ecological), and thus uneconomical in the context of the use of hydrogen for the production of fuels by CO2 hydrogenation.  For this reason, the development and testing of catalysts for processes that do not promise industrial use, is merely "art for art's sake."

We thank the reviewer for their opinion on the industrial usefulness of the proposed process. As fossil fuels deplete, global efforts have been afforded to the industrial synthesis of chemicals using readily available carbon sources, such as CO2 produced in flue gas, or sequestered in geological formations. By utilizing these CO2 sources to create valuable chemicals, it is theoretically possible to close the carbon loop on industrial chemical synthesis. This enables the production of useful carbon-based energy carriers and feedstocks, while reducing the load on the global CO2 cycle [1].

On an economic and industrial scale, it has been proven that hydrogen can be produced through water electrolysis with renewable energy input. Although it is energy intensive, excess energy, for example, from solar and wind, can be used to produce green hydrogen in an environmentally benign way [2]. The ability to transform and use this hydrogen while utilizing CO2 has been proven to be a valuable contribution to environmental and economic issues [3].

The paper is written in a vague and chaotic way. It reads and analyzes with great difficulty.

We thank the reviewer for the comment. We have revised the manuscript to ensure those details which may seem vague have been clarified. We have also rearranged parts of the discussion to make reading it clearer.

The very title of Chapter 2 Results and Discussion raises doubts as to the chronology of individual chapters. And it really is so. There is no description of the experiment yet, it is not known how the catalyst was prepared, and the authors present the results of determining its properties (2.1. Material Characterization). Throughout this chapter, where the results are presented in the form of graphs and tables, the types of analytical equipment and analysis parameters used are not given. Only Chapter 3 is the descriptions of Materials and Methods.

We thank the reviewer for the comment. The guidelines set by the Journal template dictate that the materials and methods section comes after the results and discussion. All relevant experimental parameters, analysis, and equipment are provided in Chapter 3 (Materials and Methods).

In chapter 3.2. Catalyst Characterization, the last sentence should state the concentration of HNO3 and HCl used.

We thank the reviewer for the comment. We have added the concentration of the aqua regia used to digest the samples on page 13 of the manuscript (70% HNO3 w/w and 36% w/w HCl).

Chapter 3.3.  Catalytic Apparatus and Reaction lacks the layout of the laboratory stand, where the study was conducted. The position should be schematically represented and clearly marked, where the catalyst bed is located. The rule is to provide such a description of the experiment, so that it can be repeated with the possibility to verify the results. The tests were carried out only with one set of process parameters. The experiment should be conducted under different reaction conditions (using different process parameters) to optimize the process.

We thank the reviewer for the comment. We have added a schematic diagram to show the layout of the reactor in the supporting information, and have amended the text on page 13 to reflect this.

I find subsections 2.2.1. Carbon Dioxide Conversion and Reaction Selectivity and 2.2.2. as the most valuable . Role of Potassium Substitution in Changing Reaction Selectivity. I raise no objections to these chapters. They are described correctly with a developed scientific context.

We thank the reviewer for the comment.

Reviewer 2 Report

This paper presents interesting and useful results related to carbon dioxide transformation into useful products on catalysts derived from lanthanum nickelate doped by K.  Authors applied proper characterization methods for their samples studies, discussion is reasonable, conclusions justified. Article can be accepted for publications bringing it to standards which required major revision.    

  1. English is to be polished both in sense and structure of expressions, typical examples are as follows:

Page 3, line 102  “N2 adsorption and desorption curves show a small hysteresis in the nanoparticle structure…” , hysteresis is in the amount of adsorbed nitrogen, not in the structure

Page 4, line 130: “…which is has been shown…”, etc etc.

  1. References should be brought to standard of Catalysts
  2. Procedures for preparation of samples for XPS studies are to be described in details, since for proper analysis of data for samples discharged from reactor after reduction by hydrogen or catalytic tests it is required to know whether they contacted with air being oxidized or not.
  3. Description of CO TPD profiles assigning all peaks to CO desorption from Ni metal or cationic centers (Fig. 8a) is to be more careful. It is well known that carbonyl complexes with Ni centers are easily destroyed even at room temperature by evacuation or purging with helium, etc. Hence, peak at 535 C can hardly be assigned to desorption of CO bound with Ni metal sites. FTIRS in situ spectroscopy is advised to be applied for more detailed studies of these features.
  4. Apparently for K-doped samples in reaction conditions the surface of support and  a part of Ni particles are covered by potassium hydroxycarbonate layer which is to be taken into account in discussion

Author Response

Reviewer 2

This paper presents interesting and useful results related to carbon dioxide transformation into useful products on catalysts derived from lanthanum nickelate doped by K.  Authors applied proper characterization methods for their samples studies, discussion is reasonable, conclusions justified. Article can be accepted for publications bringing it to standards which required major revision.   

We thank the reviewer for the detailed comments.

English is to be polished both in sense and structure of expressions, typical examples are as follows:

Page 3, line 102  “N2 adsorption and desorption curves show a small hysteresis in the nanoparticle structure…” , hysteresis is in the amount of adsorbed nitrogen, not in the structure

Page 4, line 130: “…which is has been shown…”, etc etc.

We thank the reviewer for the comment. We have carefully checked through the language used in the manuscript and have revised it accordingly.

References should be brought to standard of Catalysts

We thank the reviewer for the comment. We have carefully checked through the reference details and have revised it accordingly to the Journal standards.

Procedures for preparation of samples for XPS studies are to be described in details, since for proper analysis of data for samples discharged from reactor after reduction by hydrogen or catalytic tests it is required to know whether they contacted with air being oxidized or not.

We thank the reviewer for the comment. We have added this detail to the experimental details on page 12 of the manuscript in section 3.2. For the spent samples, we initially flushed the samples with N2 gas in the reactor while the samples cooled down. We then stored the samples in sealed vials purged with N2 prior to the XPS measurement so as to minimise contact with air as much as possible.

Description of CO TPD profiles assigning all peaks to CO desorption from Ni metal or cationic centers (Fig. 8a) is to be more careful. It is well known that carbonyl complexes with Ni centers are easily destroyed even at room temperature by evacuation or purging with helium, etc. Hence, peak at 535 C can hardly be assigned to desorption of CO bound with Ni metal sites. FTIRS in situ spectroscopy is advised to be applied for more detailed studies of these features.

We thank the reviewer for the comment. We do agree that carbonyl complexes with Ni centres may be removed at low temperature, however in our case, we are referring to the direct adsorption of CO on both Ni2+ (NiO) and Ni0. This phenomenon is well established in the literature. We have added a sentence to describe this effect on page 10 of the manuscript.

Apparently for K-doped samples in reaction conditions the surface of support and  a part of Ni particles are covered by potassium hydroxycarbonate layer which is to be taken into account in discussion

We thank the reviewer for the comment. We agree that under reaction conditions, the formation of potassium hydroxycarbonate is possible, and may contribute to the instability of C-intermediate products. We have added this detail on pages 10 and 12 of the manuscript.

Reviewer 3 Report

comments:

  1. Comparison of merits and demerits between methanation of CO2 and reverse water gas shift reactions by LNO perovskit with that of  transition metals and bimetallic alloy catalysts would be discussed. 
  2.  It would be ideal to show a schematic mechanism for the CO2 hydrogenation on potassium substituted LNO perovskit catalyst. 

Author Response

Reviewer 3

Comparison of merits and demerits between methanation of CO2 and reverse water gas shift reactions by LNO perovskit with that of  transition metals and bimetallic alloy catalysts would be discussed. 

We thank the reviewer for the comment. We have added discussion around selectivity comparison of the K-substituted LNO perovskite with other transition metal catalysts for CO2 reduction and the rWGS reactions on page 9 of the manuscript.

It would be ideal to show a schematic mechanism for the CO2 hydrogenation on potassium substituted LNO perovskit catalyst. 

We thank the reviewer for the comment. We have added a schematic to illustrate the proposed catalytic mechanism on the K- substituted samples in Figure S6 and amended the text on page 12 to include this.

References

  1. Hepburn, C.; Adlen, E.; Beddington, J.; Carter, E.A.; Fuss, S.; Mac Dowell, N.; Minx, J.C.; Smith, P.; Williams, C.K. The technological and economic prospects for CO2 utilization and removal. Nature 2019, 575, 87-97.
  2. Kibsgaard, J.; Chorkendorff, I. Considerations for the scaling-up of water splitting catalysts. Nature Energy 2019, 4, 430-433.
  3. Song, C. Global challenges and strategies for control, conversion and utilization of CO2 for sustainable development involving energy, catalysis, adsorption and chemical processing. Catalysis Today 2006, 115, 2-32.

Round 2

Reviewer 1 Report

Dear Authors,

The Authors completed the manuscript, which for sure enhanced its quality.

However, they did not relate at all as to the utility of such research. Therefore I renew my allegation. The Authors define the purpose of the work as "in order to increase the feasibility and applicability of utilising the catalytic conversion of CO2 into sustainable fuels and products, this work aims to tune the selectivity  of CO2 reduction between the rWGS and methanation to specific product ratios, which can be used  more efficiently for downstream processes. " At the same time, in the first sentence of Chapter 1, Introduction, they write "An economical yet effective removal and utilisation of CO2 has not been implemented on a wide scale level to reduce its impact on climate change through the greenhouse effect." It is in this statement that one should look for the purposefulness - or rather the lack of it - of conducting research such as those in the presented work. Research should support the economy and apply in industry, not being just book knowledge. Laboratory tests as a first research step are important and undeniably necessary, but their results must be able to be realistically translated into their use, and not just remain in the form of a report on the shelf. Considering the essence of the research activities, described in the paper as the possibility of using a catalyst (developed by the Authors) for CO2 reduction in the direction of CO and CO2 methanation, they are of importance only in the cognitive aspect. The basis for the implementation of process-oriented technologies, for which the Authors tested LaNiO3 perovskite catalysts through the substitution of La with K cations, is the production of hydrogen economically and environmentally justified. Hydrogen extraction in the process of water electrolysis (which is currently proposed) is highly energy-consuming (and therefore non-ecological), and thus uneconomical in the context of the use of hydrogen for the production of fuels by CO2 hydrogenation. For this reason, the development and testing of catalysts for processes that do not promise industrial use, is merely "art for art's sake."

Author Response

We thank the reviewer for their comment on the industrial usefulness of the proposed process. As fossil fuels deplete, global efforts have been afforded to the industrial synthesis of chemicals using readily available carbon sources, such as CO2 produced in flue gas, or sequestered in geological formations. By utilizing these CO2 sources to create valuable chemicals, it is theoretically possible to close the carbon loop on industrial chemical synthesis. This enables the production of useful carbon-based energy carriers and feedstocks, while reducing the load on the global CO2 cycle [1].

We agree that renewable hydrogen production is currently one of the limiting challenges of CO2 utilization as fossil-fuel derived hydrogen currently dominates the hydrogen supply chain. However, there has been a recent push to realize the economic feasibility of renewable hydrogen produced through renewable energy inputs. In this way, excess energy for example, from solar and wind, can be used to produce green hydrogen via water electrolysis [2]. Consequently, while hydrogen generation for power-to-gas systems may not be competitive at this time it will likely be so in the future such that research into materials remains relevant and is simply not art for art sake. Thus, the ability to transform and use this hydrogen while utilizing CO2 can be proven to be a valuable contribution to environmental and economic issues [3].

We have clarified this in the introduction on page 1 of the revised manuscript.

Reviewer 2 Report

Revised paper can be accepted for publication. Remaining misprints can be corrected at proofs reading stage

Author Response

We thank the reviewer for their comment.